# Functional Analysis of *StPHT1;7*, a *Solanum tuberosum* L. Phosphate Transporter Gene, in Growth and Drought Tolerance

**DOI:** 10.3390/plants9101384

**Published:** 2020-10-17

**Authors:** Minxuan Cao, Hengzhi Liu, Chao Zhang, Dongdong Wang, Xiaofang Liu, Qin Chen

**Affiliations:** 1State Key Laboratory of Crop Stress Biology for Arid Areas, College of Agronomy, Northwest A&F University, Xianning 712100, China; caominxuan@nwafu.edu.cn (M.C.); xnlhz@nwafu.edu.cn (H.L.); zhangchao520@nwafu.edu.cn (C.Z.); dongdong-1025@hotmail.com (D.W.); 2State Key Laboratory of Crop Stress Biology for Arid Areas, College of Horticulture, Northwest A&F University, Xianning 712100, China; 3College of Food Science and Engineering, Northwest A&F University, Xianning 712100, China

**Keywords:** potato, *StPHT1;7*, genetic transformation, phosphorus stress, drought stress

## Abstract

*PHT1* (phosphate transporter 1) family genes play important roles in regulating plant growth and responding to stress. However, there has been little research on the role of the *PHT1* family in potatoes. In this study, using molecular and bioinformatic approaches, 8 *PHT1* family genes were identified from the potato genome. *StPHT1;7* was highly expressed in the whole potato plants. The overexpression and silence vectors of *StPHT1;7* were constructed and transformed into the potato cultivar Desiree. Consequently, *StPHT1;7* overexpression (with a relative expression 2–7-fold that in the control) and silence lines (with a relative expression of 0.3%–1% that in the control) were obtained. Their growth vigor was ranked in the order overexpression line > wild type > silence line. In the absence of phosphorus, the root length of the overexpression line was approximately 2.6 times that of the wild type, while the root length of the silence line was approximately 0.6 times that of the wild type. Furthermore, their tolerance to drought stress was ranked as wild type > overexpression line > silence line. These results suggest that *StPHT1;7* affects growth and stress tolerance in potato plants.

## 1. Introduction

Phosphorus (P) is a macroelement required for plant growth and is one of the important elements for plant biological macromolecules, such as DNA, RNA, phospholipids, ATP, etc. P availability is also one of the main limiting factors for plant growth and development [1]. P accelerates cell division and the growth of the aboveground plant tissues. By adjusting the water content in the cell structure and the soluble sugar content in the cell, P can increase tissue osmotic pressure and enhance tolerance to drought stress. Phosphate fertilizer promotes potato root development, especially that of lateral roots, and affects root length. Plant P content is significantly positively correlated with the plant’s root dry weight [2]. Applying enough P during potato growth can significantly increase the aboveground and underground biomass of plants by 8.78% and 61.4%, respectively [3]. When P is deficient, plant growth is inhibited, red patches appear on leaves, and plant height, root length, the numbers of branches and roots, leaf area, and plant photosynthetic ability are all reduced. Most soils are not ideal for plant P absorption [4,5,6] because P can only be provided to plants in the form of inorganic P, which has a low prevalence in soils [7]. Increasing plant resource allocation to root growth can increase the roots’ acquisition of P, but an increase in root tissue will incur increased metabolic costs, and unbalanced root development will reduce the overall growth of plants [8]. Roots can consume more than 50% of the plant’s daily carbon fixation, while phosphorus-stressed plants consume more daytime net carbon assimilation than non-stressed plants [8,9]. The root system prioritizes primary growth (elongation) rather than secondary growth (radial thickening), thereby enabling the soil to absorb more phosphorus [5,6,10]. Applying P to phosphorus-deficient soil stimulates the plant’s growth response. Increasing the P supply improves the tolerances of white clover (*Trifolium repens* L.) and soybeans to dry soil conditions [11,12]. Reasons for this increased tolerance include increasing root hydraulic conductivity and maintaining leaf water potential [13], as well as increasing the number of roots deep in the soil to obtain more soil moisture [14]. Most studies have shown that plants will reduce their P uptake as soil moisture decreases [15]. Drought can prevent P absorption by reducing the distribution of P in the root system [16,17]. Moreover, phosphate fertilizer is usually used to reduce P deficiency in the soil, improve the drought tolerance of plants, and promote plant growth [18]. While the agricultural demand for phosphate fertilizers continues to grow, reserves of phosphate rock powder are rapidly declining. The best estimate for the storage life of phosphate rocks is 200 years, and the worst estimate is 50 years [19]. In practical production activities, to increase crop yields, a large amount of phosphate fertilizer is often used to alleviate the actual P deficiency of the soil, which causes a great deal of waste. At the same time, a large amount of P easily enters rivers and lakes via surface runoff, causing algal blooms.

Plants mainly take up P from the soil using the P transporters located on the cell membranes of plant roots, and the P is then transported to the plant [20,21]. Phosphorus transporter genes in plants are mainly divided into six gene families: *PHT1*, *PHT2*, *PHT3*, *PHT4*, *PHT5*, and *PHO1* [22,23,24]. Phosphate transporters from different families play different roles in plant growth and development [22,25]. According to the differences in the affinity of various transporters to P, they are divided into two transport systems: high-affinity and low-affinity [25,26,27]. The genes of the *PHT1* family are high-affinity P transporter genes. Most of these genes are expressed on the cell membranes of plant roots. The protein sequences encoded by the potato *PHT1* family are in the range of 507–535 amino acids in length, and all of these proteins have isoelectric points greater than eight, which indicates they are basic proteins. Their GRAVY values are all greater than 0, indicating that they are hydrophobic proteins [22]. These proteins are mainly responsible for the absorption and transport of P in the root system and are an important gene family for transporting and transferring P. Different members of the family have different functions. For example, in *Arabidopsis*, *AtPht1;1* and *AtPht1;4* are responsible for absorbing P in the soil, *AtPht1;5* is responsible for transporting P to the sink, and *AtPht1;8* and *AtPht1;9* are responsible for transporting and transferring P from roots [28].

Expression of the *PHT1* genes affects plant growth and development. For example, the overexpression of *OsPht1;8* slows growth and development in rice, turn the leaves yellow, and induces stress from excessive P uptake. However, growing rice under low-phosphorus conditions alleviates these symptoms, thus ameliorating P poisoning and allowing rice growth to return to normal to a certain extent. High-phosphorus and low-phosphorus conditions are not conducive to the normal growth of rice. The overexpression of *OsPht1;8* causes P poisoning and reduces the number of rice tillers and the seed setting rate. Silencing *OsPht1;8* causes severe P deficiency in the plant and further reduces the number of rice ears [29].

The *PHT1* genes also affect plants’ stress tolerance to some extent. Inserting the rice *OsPht1;2* into the soybean genome improves the ability of the soybean to absorb P and its tolerance to being under low-phosphorus stress [30]. There are similar elements in the expression of *PHT1* family genes in plants, but there are also large differences. Most members of the *PHT1* family are induced by low-phosphorus stress and are mainly expressed in the roots. There are nine *PHT1* family members in *Arabidopsis*. *AtPHT1;1*, *AtPHT1;2*, *AtPHT1;3,* and *AtPHT1;4* play important roles in P absorption [31]. *AtPHT1;5* affects the distribution of P [32]. Under long-term, low-phosphorus stress, *AtPHT1;8* and *AtPHT1;9* are expressed in the roots and play important roles in translocating P from the roots to the aboveground organs. The rice *PHT1* family contains 13 genes [33]. Under high phosphorus conditions, P absorption and transport depend on *OsPHT1* [34], whereas under low-P stress, gene overexpression lines of *OsPHT9* and *OsPHT10*, which are expressed in the roots, significantly increase P uptake [35].

Potatoes are an indispensable food crop across the world and the fourth largest food crop in China. However, the arid climate and phosphorus-deficient soil in northwest China is not suitable for potato cultivation. At the same time, with the increasing frequency of extreme weather around the world, climate has become unstable and droughts in certain areas may become more frequent [36,37]. There are no reports of direct cloning of *PHT1* family genes in potato. Therefore, it is necessary to study the *PHT1* family with regard to its ability to increase P absorption and improve crop quality in potatoes, which is of positive significance for increasing agricultural output in arid and phosphorus-deficient areas. In this study, using bioinformatics and molecular biology methods, eight *PHT1* genes were identified in the potato genome, and *StPHT1;7* was cloned into potato for the first time. Overexpression and interference techniques were used to explore the effects of *StPHT1;7* on potato self-development and stress tolerance. Further research and analysis of the function of the potato *PHT1* family of P transporters on plant growth and tolerance to stress will help lay an important theoretical basis for the cultivation of high-phosphorus and stress-resistant potato varieties.

## 2. Results

### 2.1. Gene Structure, Conservative Motifs, and Multiple Sequence Alignment of the Potato PHT1 Gene Family

Eight *PHT1* family genes were retrieved from the International Potato Genome Sequencing Consortium (PGSC) database and the gene ID for each is shown in Appendix A. The results of cluster analysis (Figure 1A) showed that similar cluster relations have similar gene structures. For example, both *StPHT1;1* and *StPHT1;2* only have one exon and identical motifs. Gene structure analysis (Figure 1B) showed that the potato *PHT1* genes have a simple structure and no more than one intron. *StPHT1;8* contains one intron sequence and two exons, while the other genes only contain one exon. In total, eight conserved domains were detected in the amino acid sequences, among which *StPHT1;5* and *StPHT1;8* lacked one domain each, while the remaining genes all contained these eight conserved domains (Figure 1C).

By blasting the eight potato PHT1 proteins (Figure 2), all of the protein sequences had [ST]-x-[RK] motif, N-{P}-[ST]-{P} motif, and [ST]-x(2)-[DE] motif. Putative phosphorylation sites are present as protein kinase C phosphorylation sites at positions 239 to 241 displaying the [ST]-x-[RK] consensus motif and at positions 498 to 501 as casein kinase II phosphorylation sites after the [ST]-x(2)-[DE] consensus motif (numbers refer to the amino acid sequence of the StPHT1;3 protein as shown in Figure 2). In the [ST]-x-[RK] motif, the second amino acid is different. StPHT1;3 and StPHT1;6 are glycine while the other proteins are alanine. In the [ST]-x(2)-[DE] motif, all potato PHT1 amino acid sequences are identical. An N-glycosylation site is present at positions 420 to 423 displaying the consensus motif N-{P}-[ST]-{P}. In the N-{P}-[ST]-{P} motif, the second amino acid is different. StPHT1;3 and StPHT1;6 are serine, StPHT1;8 is threonine, and the other proteins are alanine. The first four motifs with the lowest motif E-value in the Figure 1C legend are also shown in Figure 2. There is a conserved amino acid sequence (GGDYPLSATIMSE) in all the PHT1 family genes. It suggested that these amino acids might have some special biological functions in potato plants.

### 2.2. The Phylogenetic Tree of Potato PHT1 Genes

Compared with tomato, *Arabidopsis*, and soybean, each branch of the potato *PHT1* family protein phylogenetic tree has a high bootstrap value, and the potato *PHT1*-encoded proteins can be divided into three phylogenetically defined groups. According to their clustering, StPHT1;1, StPHT1;2, StPHT1;4, StPHT1;5, and StPHT1;7 are classified into group I; StPHT1;3 and StPHT1;6 are classified into group II; and StPHT1;8 is classified into group III (Figure 3). These differences in protein structure among the potato PHT1 family P transporters may be related to their different roles in the process of P transport within the plant.

### 2.3. Potato PHT1 Protein Is Located on the Cell Membrane

By analyzing their primary structure, we found that the proteins encoded by the potato *PHT1* family have transmembrane domains (Figure 4). The N-termini are all located in the cytoplasm. Potato *PHT1* group I proteins (StPHT1;1, StPHT1;2, StPHT1;4, StPHT1;5, StPHT1;7) contain 10 transmembrane structures, except for StPHT1;5, which contains 9 transmembrane structures. For the group 1 proteins, the C-termini are all located inside the cell, except for the extracellular C-terminus of StPHT1;5. Group II (StPHT1;3 and StPHT1;6) and group III (StPHT1;8) proteins contain either 11 or 12 transmembrane structures, for which all of the C-termini are extracellular. None of the eight PHT1 family members have a signal peptide sequence. This shows that the potato *PHT1* family genes encode P transporters located on the cell membrane.

### 2.4. Potato PHT1 Family Members Have Similar Protein Secondary Structure

The protein secondary structures are generally similar among the eight members of the potato *PHT1* family (Figure 5). The most common feature of their secondary structures is the alpha-helix (Appendix A). This is followed by random coils and extended strand structures. Beta-turns are the least common secondary structure found within the potato *PHT1* family proteins, as their highest content, in the StPHT1;7 polypeptide chain, is only 5.02%. The secondary structure of transmembrane domain is mostly alpha-helix.

### 2.5. Expression Patterns of the Potato PHT1 Gene

As shown in Figure 6A, the expression of different potato *PHT1* genes appears to have different tissue specificity. *StPHT1;7* was expressed at very high levels in all plant tissues examined, whereas the expression of all other genes was low in two or more of the tissues examined. Many of the genes were not expressed (expression level = 0) in some tissues, including *StPHT1;3*, whose expression was difficult to detect, except for a small amount of expression at the level of the whole plant.

As shown in Figure 6B, the expression changes of each gene are different under different stress treatments. For example, under heat stress, the expression of *StPHT1;5* remains unchanged, while the expression of *StPHT1;3* and *StPHT1;6* decrease, and the expression of the remaining five genes all increase. Under mannitol stress, the expression of *StPHT1;3* and *StPHT1;6* remain unchanged, while the expression of *StPHT1;4* and *StPHT1;7* decrease and the expression of the other four genes increase. Under NaCl stress, the expression of *StPHT1;4*, *StPHT1;5*, and *StPHT1;7* decrease, and the expression of the other five genes increase. *StPHT1;7* expression increased under heat stress but decreased under mannitol and NaCl stress.

Combining the results shown in Figure 6A and Figure 6B, it can be seen that *StPHT1;7* is highly expressed across all parts of the potato plant and that its expression level changes after being subjected to heat stress, mannitol stress, and NaCl stress. From this, we infer that *StPHT1;7* plays an important role in P transport in potato plants and in response to stresses. Therefore, *StPHT1;7* was selected for further research on its function in potato growth and stress tolerance.

### 2.6. Silencing and Overexpression of StPHT1;7

The results showed that the transgenic plants contained the NPTII gene (676 bp) inserts, whereas the negative control (WT) did not (Figure 7). We ultimately identified eight silence and eight overexpression transgenic plants.

The expression levels of all the lines can be seen in Figure 7A,B. To further investigate the underlying functions of *StPHT1;7* in potato, we selected some representative transgenic lines for the following experiment. The silence lines are *StPHT1;7* RNAi #1, *StPHT1;7* RNAi #2, *StPHT1;7* RNAi #7, and *StPHT1;7* RNAi #11, and the overexpression lines are *StPHT1;7* OE #3, *StPHT1;7* OE #8, *StPHT1;7* OE #9, *StPHT1;7* OE #12, *StPHT1;7* OE #18, and *StPHT1;7* OE #21.

### 2.7. Phenotypic Characteristics of Transgenic Plants

Under normal culture conditions, the phenotypes of the overexpression lines were similar to each other and the phenotypes of the silence lines were similar to each other, except for that of *StPHT1;7* RNAi #11. Therefore, some representative lines were selected for display in Figure 8. The phenotype of *StPHT1;7* RNAi #11 was different from the other silence lines only 35 days after transplanting, but its growth and development status were the same at other stages. As shown in Figure 8A, 35 days after transplanting, the plant heights of the overexpression plants were higher than those of the silence and WT plants. Silence plants had many branches and side branches and the leaves were elongated. In addition, *StPHT1;7* RNAi #11 showed leaf deformities and multi-branched phenotypes later than other silence lines.

As shown in Figure 8B, the growth of the overexpression plants was stronger than that of the silence and WT plants, with larger and a greater number of leaves and thicker stems. A no flowering phenomenon was observed in all silence plants. Further, their growth and development were slow. As shown in Figure 8C and D, the root systems of the overexpression plants were the most developed. The stem hairs of the silence lines were long and straight with less curving, while the stem hairs of the overexpression lines and WT were short and curved (Figure 8E). In conclusion, *StPHT1;7* has an important effect on potato growth and development.

### 2.8. Responses of the Transgenic Plant to Phosphorus Stress

It can be seen in Figure 9A that the plant height, root length, and leaf area of the overexpression lines under normal culture conditions were all greater than those of the WT, and the indicators of the silence line were less than those of the WT. Under phosphorus-free conditions, the height and leaf area of each plant decreased compared to the values under normal phosphorus (1mM) conditions, but the index of the overexpression plant was still greater than that of the WT, and the silence plants was the smallest. It can be seen in Figure 9B that under normal culture conditions, the leaf phenotypes of the overexpression and WT lines were normal. The leaves of the silence line were elongated, but the appearance of deformities in the leaves of *StPHT1;7* RNAi #11 was slower than that of other silence lines. Under phosphorus-free conditions, none of the leaves of the silence lines were deformed, but the leaf sizes were much smaller than those of the overexpression and WT lines. As shown in Figure 9C, after 30 days of Hoagland hydroponics, overexpression plants exhibited the highest root growth and root development, followed by WT, with the root growth and development of the silence plants being the weakest. At the same time, under P stress, the root system of overexpression lines was slightly longer than that of the normal culture. Regardless of whether there was P deficiency, the root growth of *StPHT1;7* RNAi #11 was better than that of *StPHT1;7* RNAi #2, and the expression level of the *StPHT1;7* RNAi #11 line was about 10 times that of the *StPHT1;7* RNAi #2 line, which further explains the effects of *StPHT1;7* on potato root development.

The differences in plant height under normal P conditions can be seen in Figure 9D. On the 15th day of the culture, the plant heights of the overexpression lines were significantly higher than those of the silence lines. On the 30th day, the difference in plant height between the overexpression lines and the silence lines further increased compared with the 15th day, and compared with the WT; plant heights of the overexpression lines were significantly higher.

The differences in plant height under P-free stress can be seen in Figure 9E. The plant heights of the overexpression lines were significantly higher than those of the silence lines and the WT on the 15th day of culture. However, by the 30th day in culture, the plant heights of the overexpression and WT plants were not significantly different.

The difference in root length under normal P conditions can be seen in Figure 9F. On the 15th day of culture, the root lengths of the overexpression lines were significantly greater than those of the silence and WT lines. The WT root length was significantly longer than that of the silence lines. On the 30th day, the order was the same as that on the 15th day, but the difference in root length between the silence lines and WT was not significant.

The difference in root length under phosphorus-free stress can be seen in Figure 9G. On the 15th day of culture, the root lengths of the overexpression lines were significantly higher than those of the silence lines and WT, but the difference in root length between the WT and the silence lines was not significant. By the 30th day, the difference in root length between the overexpression lines and the silence lines and WT had further increased.

### 2.9. Response of Transgenic Plants to Dehydration Treatment

As the concentration of PEG6000 increases, the water potential in the medium decreases, and the roots of the potato tissue-cultured seedlings became unable to normally obtain the water needed for growth, which affects the growth of the potato. Moreover, the phenotype is different (Appendix A and Figure 10).

In the treatment without PEG6000, the root lengths, leaf areas, and plant heights of the overexpression and silence lines show certain differences because the expression levels of *StPHT1;7* were different between different lines, and the absolute values of some measured indexes were not comparable. Therefore, the absolute value of the index of each line can be converted into a relative value for comparison, which can eliminate the errors caused by the internal differences between the lines to a certain extent. The data were converted into relative growth indicators for the subsequent difference analysis (Appendix A).

Comparing the growth parameters of the potato tissue culture seedlings under 15% and 20% PEG6000 stress, it was found that under 20% PEG6000 stress, although the coefficient of variation for each index was roughly the same as the values under 15% PEG6000 stress, most of the growth parameters reached their lowest values under 20% PEG6000 stress, and some were no longer comparable. Therefore, the 15% PEG6000 data were used to further analyze the drought tolerance of the potato plants. Multiple comparisons of the relative growth indicators for potato lines at 15% PEG6000 concentration are shown in Table 1.
Relative Growth Parameter = Growth Parameter (PEG concentration 15%)/Growth Parameter (PEG concentration 0%) × 100%

According to the coefficient of variation (CV) in Table 1, the relative leaf area was the index most strongly affected by PEG6000 stress and the relative plant height was the least affected by PEG6000. The CV of root indexes was larger than that of stem indexes, indicating that the roots were more susceptible to drought stress than the stems. According to all of the indexes of the five potato lines, the WT performed best. Except for relative root length, all other indicators for WT were significantly higher or at least not significantly different from the other lines. This shows that the WT was less affected by PEG6000 stress and had strong drought tolerance. Each of the remaining four lines had multiple indicators that were lower or significantly lower than those in the other lines, so the drought tolerance of the remaining four lines could not be judged here. In summary, drought tolerance followed the order of WT > (*StPHT1;7* RNAi #1, *StPHT1;7* RNAi #7, *StPHT1;7* OE #8, *StPHT1;7* OE #21 in random order).

### 2.10. Response of Transgenic Plants to Drought Stress

As shown in Figure 11A, on the 23rd day of drought stress, the whole plant of the overexpression line had wilting symptoms or the uppermost leaves showed obvious symptoms of water loss. However, the silence line had already dried up. The test results show that the drought tolerance of the overexpression, silence, and WT potato plants from highest to least tolerance was WT > overexpression line > silence line. The result of the previous PEG6000 stress test showing that the WT had the strongest drought tolerance was verified.

Figure 11B shows that during the experiment, the soil water content of the five potato lines and the blank control gradually decreased with an extension of the drought stress time, but this decline was not large or consistent over the 0–14 day period. The decline in the following 14–23 days was large, and the difference was extremely significant. For drought stress on the 7th and 14th days, although the difference in soil water content between the five potato lines and the control was not large, significant differences appeared. On the 23rd day of drought stress, the specific values of the soil water content of the overexpression lines, silence lines, and WT lines showed great differences. The average soil water content of overexpression lines reached 1.41%, which was the lowest value. The average soil water content of the silence lines was 23.29% and was the highest in pots containing potato plants.

Figure 11C shows the leaf water content on the 23rd day of drought treatment. The leaf water content of the silence lines was lower than 20% and had dried up. The leaf water content of overexpression lines was lower than that of WT, and obvious wilting symptoms appeared. Figure 11D,E shows the plant height and root length on the 23rd day of drought stress. The overexpression lines had the fastest growth rate so that they consumed more water in soil, which led to wilting earlier than WT. The growth rate of silence lines was the slowest, however, when the soil water content was still higher than 20%, the silence lines had already seriously withered and dried up, indicating that their drought tolerance ability was the weakest.

## 3. Discussion

### 3.1. Bioinformatics Analysis of Potato PHT1 Family

In plants for which the *PHT1* family has been characterized, there are 12 members (genes) in rice, 9 in *Arabidopsis*, 8 in barley, and 4 in maize [31,33,40]. Studies have shown that the evolution of the potato was caused by gene family amplification and tissue-specific expression [41]. However, there are few reports on cloning the *PHT1* family in potato at the genome level. In this study, by using bioinformatics methods to analyze the *PHT1* members in the reported plants, eight *PHT1* genes have been discovered in the potato genome. This indicates that the *PHT1* family retained a largely fixed function of the genetic evolution of different species. Previous studies on *PHT1* family genes in rice and ryegrass were focused on the genes with high expression in tissues and altered expression under stress [40,42]. Therefore, we chose *StPHT1;7* with a high expression level across the various tissues (Figure 6A) for further research.

### 3.2. StPHT1;7 Expression Affected Potato Plants Growth

Different *StPHT1;7* expression levels affected potato physiological indicators and stress tolerance. In the BETA (choline dehydrogenase gene) transgenic *Populus nigra* plants, different transgenic lines of different BETA expression levels have different tolerance to salt stress [43,44]. In our study, the expression levels varied greatly among the silence lines (Figure 7A). Notably, *StPHT1;7* RNAi #11 line’s root lengths (Figure 9C,G) and shape of leaves (Figure 9B) were different from the other silence lines. This indicates that different gene silencing efficiency has different effects on potato plants. This study also found an interesting phenomenon. The stem hair of the silence plant was evidently longer than that of the control (Figure 8E). There has been little research on potato stem hair. Therefore, the results of this study can provide reference for further study.

### 3.3. Phosphorus Stress Had Different Effects on Transgenic Plants

The main way that plants adapt to low-phosphorus environments is by expanding the surface area of their root systems, which is represented by an increase in the total length of the root system, the number of lateral roots and root hairs, and the density [45]. At the same time, plants with sufficient P and strong P absorption capacity are also more root-developed [46]. In this study, when the transgenic plants were subjected to P stress, some of our results were consistent with these expectations. The root systems of the overexpression and silence plants were different (Figure 9C), reflecting the strength of the P absorption capacity. The stronger this ability was, the longer the root system grew [46]. The silence lines behaved differently, which is consistent with the previous conclusions when they were cultured for 15 days without P [45]. However, at 30 days, the root indexes of the phosphorus-free culture were weaker than those of the normal culture. This finding was not in accordance with previous studies. This phenomenon possibly is because the growth of the silence plant itself was weak and the lack of nutrient cultivation further weakened its growth, resulting in a decrease of the root indexes.

### 3.4. StPHT1;7 Affect Potato Plants Drought Tolerance

With an increase in the PEG concentration, most of the growth indices for potatoes decreased [47]. The early stages of drought stimulate the development of plant roots and increase the root-to-shoot ratio [48]; part of the indicators increase first and then decrease with an increase in PEG stress. In this study, root length generally reaches a peak at about 10% PEG concentration, and leaf area reaches a maximum when the PEG concentration is about 5%. The PEG concentration at the turning point of the indicators obtained in this experiment was different from that found in previous studies. Lu et al. showed that root length is significantly inhibited at a 4% PEG concentration [49]. When transplanting tissue culture seedlings under drought stress, the tissue culture seedlings in our study were transplanted with roots after 20 days of pre-cultivation, which is different from the methods used by Lu et al. [49] (directly transplanting cut stems without roots), so the survival rate and the overall growth in our study were higher, which resulted in an increased tolerance of the plants to drought. At the same time, the trends for changes in physiological indicators were more obvious. This indicates that roots play an important role in plant growth and drought tolerance. Therefore, in PEG stress experiments, potato plants with roots can distinguish drought tolerance better than potato plants consisting of cut stems without roots.

According to our PEG drought stress results, the WT was the most drought tolerant. It may be that the overexpressing and silencing of *StPHT1;7* affected potato plant growth and development, stomatal opening and closing, antioxidant capacity, enzyme activity, and metabolism [1], which in turn affected drought tolerance. In the following natural drought stress test, WT was still the most drought tolerant, and the drought tolerance of the overexpression lines was obviously stronger than that of the silence lines. These results further validated previous studies on some *PHT1′*s response to drought [50,51]. Looking back to growth development in PEG treatment, the relative root length index of *StPHT1;7* OE #21 was as low as 53.38%, while that of *StPHT1;7* RNAi #7 was as high as 94.94% (Table 1). However, the absolute root length of *StPHT1;7* OE #21 was still 67.3 mm and that of *StPHT1;7* RNAi #7 was only 31.6 mm. Therefore, when analyzing the strength of drought tolerance, it is necessary to comprehensively consider both relative indicators and absolute indicators to avoid inaccurate results.

In our study, physiological indicators reflected the differences in growth and stress tolerance among potato plants. The well-developed aboveground parts and root systems of potato plants are conducive to photosynthesis and nutrient absorption, which is conducive to the growth of underground tubers. The influence of *StPHT1;7* has reference significance for potato breeding. Breeding plants that highly express *StPHT1;7* could enhance the plant’s phosphate fertilizer utilization efficiency, which would be conducive to potato growth and drought tolerance, increasing potato production, and guaranteeing world food security.

## 4. Materials and Methods

### 4.1. Identification of StPHT1s and Analysis of Their Physical and Chemical Parameters

According to Liu et al.’s naming rules and sequence numbers for the *PHT1* gene family [22], the PGSC database (http://solanaceae.plantbiology.msu.edu/) was used to search for the *PHT1* family gene sequences, and ProtParam (https://web.expasy.org/protparam/) was used to predict the physical and chemical properties of the corresponding proteins. Plant-mPloc 2.0 (http://www.csbio.sjtu.edu.cn/bioinf/plant-multi/) was used to predict where the protein would play a role in the cell.

### 4.2. Construction of Phylogenetic Tree of PHT1 Gene Family

Clustal W was used for multiple sequence alignment of *PHT1* family protein sequences. Then, we used MEGA7.0 software to construct a phylogenetic tree (built using the neighbor-joining method, setting the bootstrap value to 1000). Finally, we used evolview software (https://www.evolgenius.info/evolview/#login) to beautify the phylogenetic tree.

### 4.3. Analysis of Potato PHT1 Gene Structures and Conserved Domains

We downloaded the CDS sequence of the potato *PHT1* gene family from the PGSC database. The Gene Structure Display Server 2.0 (http://gsds.cbi.pku.edu.cn/) was used to analyze the intron and exon regions of each *PHT1* family gene sequence, and MEME (Version 5.1.1) (http://meme-suite.org/tools/meme) was used to analyze the conserved regions of the protein sequences of the potato *PHT1* gene family. Clustal X 1.83 software was used for multiple sequence alignment of *PHT1* family protein sequences.

### 4.4. Analysis of the Primary and Spatial Structure of Potato PHT1 Proteins

MEMSAT-SVM (http://bioinf.cs.ucl.ac.uk/psipred/) was used to predict the transmembrane domains of the *PHT1* family genes. TOPCONS (http://topcons.cbr.su.se/) was used to predict the protein signal peptides. Using the potato *PHT1* gene family protein sequences, SOPMA (https://npsa-prabi.ibcp.fr/cgi-bin/npsa_automat.pl?page=npsa_sopma.html) and PSIPRED 4.0 (http://bioinf.cs.ucl.ac.uk/psipred/) were used to predict the secondary protein structures of the proteins encoded by the potato *PHT1* gene family.

### 4.5. Tissue Expression and Stress Treatment Expression Analysis of the Potato PHT1 Genes

We downloaded the potato *PHT1* family gene fragments per kilobase per million (FPKM) values from the PGSC database, selected the tissue-specific expression levels of the heterozygous diploid breeding line (RH) and the stress expression level of the doubled monoploid potato (DM), and developed two expression heat maps using HEMI (Version 1.0) software (http://hemi.biocuckoo.org/).

### 4.6. Growth Status of Plant Materials

Potato Desiree tissue culture seedlings were stored in a plant incubator in our laboratory and grown in an MS medium containing 20 g sucrose, with a light/dark time of 16 h/8 h and a light intensity of 12,000 Lx; for all potato tissue culture seedlings stored in the plant incubator in this experiment, the light and temperature conditions were the same.

### 4.7. Silence and Overexpression of StPHT1;7

We selected the pBIN19 and pBI121 vectors for reconstructing the silence and overexpression vectors. We used the gateway method to construct the silence recombination vector, selected 451 bp as the interference fragment in the conservative segment of the *StPHT1;7*, designed specific primers for the interference fragment (StPHT1;7i-F: GGGGACAAGTTTGTACAAAAAAGCAGGCTCTATGACAATGCATTGAAACCT; StPHT1;7i-R: GGGGACCACTTTGTACAAGAAAGCTGGGTAGAAGCAAGAGCATCAACCTC), and used KOD-Plus-Neo (Toyobo, Osaka, Japan) to perform PCR amplification to obtain the interference fragments. The pDONR201 entry vector of the interference fragment was constructed via the BP reaction and was named pDONR201-StPHT1;7i after successful sequencing. The pBIN19 expression vector of the interference fragment was constructed via the LR reaction and was named pBIN19-StPHT1;7i after successful sequencing; it was introduced into the LBA4404 *Agrobacterium* (Mei5, Beijing, China) using the freeze-thaw method. The transformation of transgenic *Agrobacterium* into Desiree leaves was conducted following the methods described by Molla et al. [52] with minor modifications. Our transformation media, which differed from that of Molla et al. [52], is described in Table 2. The transgenic plants were identified using PCR with vector-specific primers, and the transgenic plants with good root growth on the kanamycin resistant medium were selected for DNA extraction. The DNA of the non-transgenic Desiree plant was used as a negative control, and the neomycin phosphotransferase (NPTII) on the vector was used to design the gene primers (NPTII-F: 5′-GCTATGACTGGGCACAATCAG-3′; NPTII-R: 5′-ATACCGTAAAGCACGATTGAA-3′) used for PCR detection. The expected amplified fragment size was 676 bp. Eight plants with positive PCR results were selected for their RNA to be extracted. Total RNA was isolated from transgenic and wild-type (WT) potato leaves using the RNAsimple Total RNA Extraction Kit (Tiangen, Beijing, China). First-strand cDNA was synthesized from 500 ng of total RNA using the FastKing RT Kit (with gDNase) (Tiangen, Beijing, China). The cDNA was diluted five-fold with nuclease-free water, and all procedures were performed according to the instructions. Quantitative real-time (qRT)-PCR was performed on a Bio-Rad analysis system (CFX96, Hercules, CA, USA) using SYBR Green PCR Master Mix (Tiangen, Beijing, China). Ubi3 was used as a reference gene (ubi3-F: TCCGACACCATCGACAATGT; ubi3-R: CGACCATCCTCAAGCTGCTT). Specific quantitative primers (RTStPHT1;7-F: CTCAGGCTGATTTCGTGTGG; RTStPHT1;7-R: TTCGACTTGCAACACCTTGG) were used to analyze the relative gene expression. The qRT-PCR program was set to 5 min at 95 °C; followed by 40 cycles of 15 s at 95 °C, 15 s at 62 °C, and 30 s at 72 °C; and 72 °C for 10 min. The relative expression level of the *StPHT1;7* was evaluated using the 2^−△△Ct^ method [53].

The target gene sequence primers were designed. Based on the XbaI and SacI enzyme-digested pBI121 vector, the enzyme digestion site and vector homologous sequence were inserted into the primer (StPHT1;7-OE-F: CACGGGGGACTCTAGATAGGCGAACGATTTGCAAGT; StPHT1;7-OE-R: GATCGGGGAAATTCGAGCTCTTAAACAGGAACTGTCCTTC). To obtain the target fragment of *StPHT1;7*, PCR was performed using the cDNA from Desiree as the template and KOD-Plus-Neo (Toyobo, Osaka, Japan) as the DNA polymerase. A one-step cloning kit was used to obtain the recombinant vector, which was named pBI121-StPHT1;7 after successful sequencing and then introduced into *Agrobacterium* LBA4404 using the freeze-thaw method. The transformed plants were obtained as previously described. Once DNA identification was verified, eight plants were selected for qRT-PCR identification using the same method as described above.

### 4.8. Phenotype Identification of the Transgenic Potato

We transplanted the transgenic tissue culture seedlings into small pots with vermiculite and cultivated them in a greenhouse under a temperature of 22 ± 1 °C, a light/dark time of 16 h/8 h, and a light intensity of 12,000 Lx. We added Hoagland nutrient solution once every 5 days. After 40 days, the plants were transplanted into large pots containing nutrient soil, and tap water was supplied once every 5 days. After 3 months, the plants were photographed and recorded.

To identify the root system of transgenic potato, we selected the tissue-cultured seedlings that had been cultivated for 30 days, excised the shoot tips of the tissue-cultured seedlings on an ultra-clean bench, and planted those shoot tips in MS2 liquid medium. Each line was planted in 3 bottles, and 3 plants were evenly placed in each bottle and then cultivated in an incubator. After 25 days in the MS liquid culture, the roots of the tissue-cultured seedlings were cut off, and the roots were scanned with an LA-S root scanner.

### 4.9. Transgenic Plant Phosphorus Stress Treatment

For this experiment, we used P (1 mM) and P-free nutrition for the control and treatment, respectively. For the control treatment, the potato nutrient solution used in this experiment was Hoagland nutrient solution, formulated according to Epstein’s formula [54]. The P-free Hoagland formula used SO_4_^2−^ instead of PO_4_^3−^, and the concentration of N, P, and K was the same as the total nutrients in the Hoagland nutrient solution.

We took 40-day-grown potato transgenic seedlings, selected nine replications from each line, cut off the roots, and developed a hydroponic culture cultivated in a greenhouse under a temperature of 22 ± 1 °C, a light/dark time of 16 h/8 h, and a light intensity of 12,000 Lx. We replaced the nutrient solutions every 5 days. Physiological indicators were measured on the 0th, 15th, and 30th days.

### 4.10. Transgenic Plant PEG6000 Stress Treatment

The preparation method for the polyethylene glycol (PEG) medium was based on the method for pouring PEG plates used by the Zhu JK Laboratory [55]. PEG8000 was changed to PEG6000, and the standing time of the PEG liquid on MS2 was changed to 48 h. We prepared the medium with PEG6000 concentrations of 0, 5%, 10%, 15%, and 20% in the medium. For the 0 PEG solid medium (the control), we used 1/2MS liquid without PEG instead of liquid containing PEG.

Five lines were selected and cultured in tissue culture bottles for 20 days with consistent growth of tissue culture seedlings, including two overexpression lines: *StPHT1;7* OE #8 and *StPHT1;7* OE #21; two silence lines: *StPHT1;7* RNAi #2 and *StPHT1;7* RNAi #7; and one WT. On the ultra-clean bench, all the roots of the seedlings were transferred to the medium containing PEG6000 of different concentrations. Each line was planted in three bottles for each PEG concentration, and five plants were evenly planted in each bottle, so each line had 15 replications. After transplanting, the tissue culture seedlings were placed in an incubator for cultivation. After 25 days of PEG6000 water stress, we took pictures and measured physiological indicators. The conversion formula in Table 1 is shown in Formula 1 below.
Relative Growth Parameter = Growth Parameter (PEG concentration 15%)/Growth Parameter (PEG concentration 0%) × 100%(1)

### 4.11. Transgenic Plant Drought Stress Treatment

Tissue-cultured seedlings with consistent growth cultured in a normal MS2 medium for 40 days were selected, transplanted into small pots with vermiculite, and cultivated in a greenhouse. We added Hoagland total nutrient solution every 3 days and cultivated the plants for 30 days.

Starting from the 30th day, nine lines were selected as the four silence lines (*StPHT1;7* RNAi #1, *StPHT1;7* RNAi #2, *StPHT1;7* RNAi #7, and *StPHT1;7* RNAi #11) and four overexpression lines (*StPHT1;7* OE #3, *StPHT1;7* OE #8, *StPHT1;7* OE #9, and *StPHT1;7* OE #12), while one was selected as the control WT. We selected 15 plants that exhibited similar seedling growth state under drought stress. On day 0, we soaked each pot with Hoagland solution until the maximum soil water content was reached. We then added the blank control (12 pots with the same pot and fresh new vermiculite) and soaked each pot with Hoagland solution until reaching the maximum water content of the soil. The soil water content and leaf water content were measured at 10:00 a.m. on the 0th, 7th, 14th, and 23rd days, and the physiological changes brought about by drought stress were observed. We selected 3 pots from each line to determine the various indicators.

### 4.12. Determination of Relevant Physiological Indicators

The following 4 indicators were measured with vernier calipers: plant height (from the lowest part of the stem base to the shoot tip of the potato), stem diameter (the thickest part of the bottom of the stem had two horizontal and vertical values, and we took the average), root length (the length of the longest root), and leaf area (we measured the length and width of the larger of the second and third leaves with the top fully expanded—leaf area = 0.76 × length × width). The following 6 indicators were measured with an electronic balance: the fresh weight of stems and roots (we weighed the fresh weight of the stems and roots with an analytical balance after sampling) and the dry weight of stems and roots (we dried the stems and roots at 65 °C for more than 72 h to a constant weight and then weighed them), the water contents of the soil and leaves (we weighed the fresh weights of the soil and leaves, dried them at 65 °C for more than 72 h to a constant weight, weighed the dry weights of the soil and leaves, and calculated the moisture content). All root systems had root counts greater than 5 mm, including the aboveground roots. When photographing the stem hairs, we selected the stems between the third and fourth fully expanded leaves of the potato plant from the top to the bottom and took a photo with a dissecting microscope. All data analyses were performed using SPSS 22.0 and excel.

## 5. Conclusions

In this study, eight members of the potato *PHT1* family were identified. *StPHT1;7* is located on chromosome 9 and has 10 transmembrane structures. Among the members of the *PHT1* family, *StPHT1;7′*s expression levels were the highest across potato tissues. Its expression level was obviously changed under stress and was predicted to play an important role in P transport and stress tolerance in potatoes. *StPHT1;7* expression increased in *StPHT1;7* overexpression lines, which promoted growth and development of the potato plants, causing them to exhibit greater plant height and, especially, longer roots. *StPHT1;7* expression decreased in *StPHT1;7* RNAi-silenced lines, which hindered the growth of the potato plants to a certain extent, resulting in slow growth and root lengths that were significantly lower than those of the WT. Under phosphorus-free conditions, root length in the overexpression lines increased, becoming significantly longer than those of the WT and silence lines. The plant heights of the overexpression line still exceeded those of the silence lines and the WT under phosphorus-free conditions, but this difference was reduced compared to that under normal culture. Under drought stress, both the overexpression lines and the silence lines were inferior to the WT lines in drought tolerance, and the silence lines exhibited the least drought tolerance. These results provide additional evidence that *PHT1* family genes function during drought stress and lay a foundation for studying the molecular mechanisms underlying drought response in potatoes. Moreover, these findings will serve as a reference for future studies on the responses of potatoes and other crops to drought stress and phosphorus-free stress. In subsequent studies, we will further explore the impact of *StPHT1;7* on potato plants and improve stress tolerance in potato plants by regulating *StPHT1;7*.

## Figures and Tables

**Figure 1 plants-09-01384-f001:**
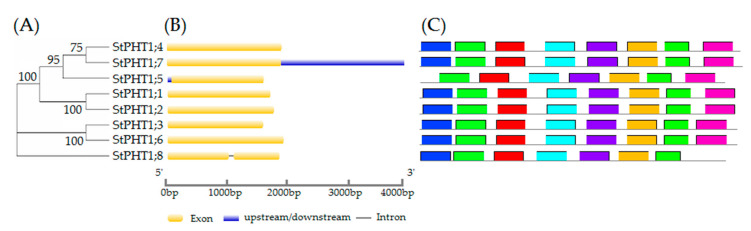
(**A**) Phylogenetic tree of the potato *PHT1* gene protein. The potato PHT1 amino acid sequences were aligned using Clustal W. The phylogenetic tree was constructed using the neighbor-joining method, setting the bootstrap value to 1000. The numbers on the branch represent the bootstrap value. (**B**) Gene structure of the potato *PHT1* gene family. The yellow boxes indicate the coding sequences. The discontinuous lines indicate the introns of these genes. The blue boxes indicate the upstream/downstream. (**C**) Domain prediction of the potato PHT1 proteins. The right side represents the motif composition associated with each StPHT1 protein. The motifs are displayed in different colored boxes. Motifs are sorted by E-value from small to large; different colors correspond to the motifs of the same color in Figure 1C.

**Figure 2 plants-09-01384-f002:**
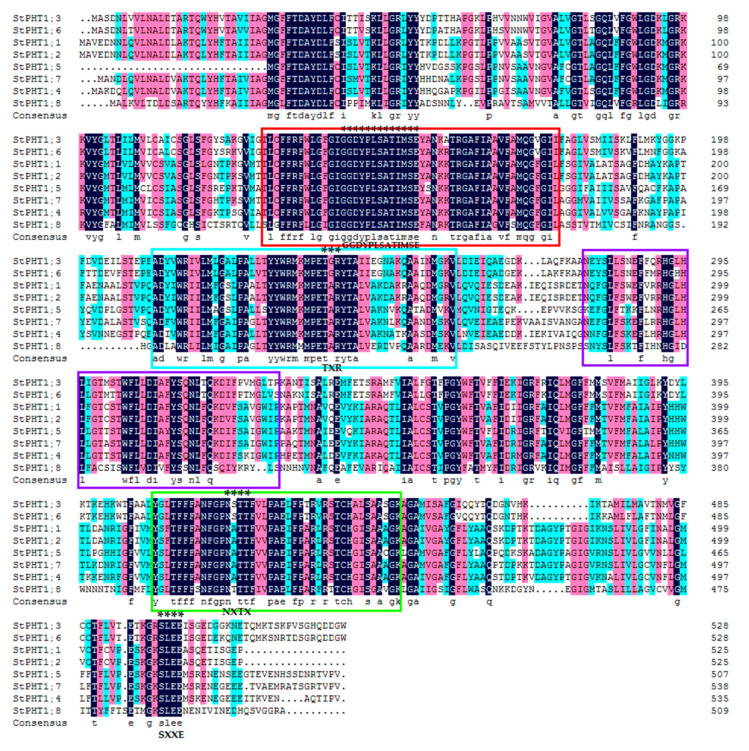
Potato PHT1 family protein sequence alignment. The amino acid sequence of the potato PHT1 protein was aligned using Clustal X 1.83. Amino acids marked in dark blue indicate 100% sequence identity, pink indicates ≥75% identity, and light blue indicates ≥50% identity. The boxes of different colors correspond to the motifs of the same color in Figure 1C. Asterisks (*) above the sequence indicate the [ST]-x-[RK] motif, N-{P}-[ST]-{P} motif, [ST]-x(2)-[DE] motif [38], and conserved amino acid sequence (GGDYPLSATIMSE) [39].

**Figure 3 plants-09-01384-f003:**
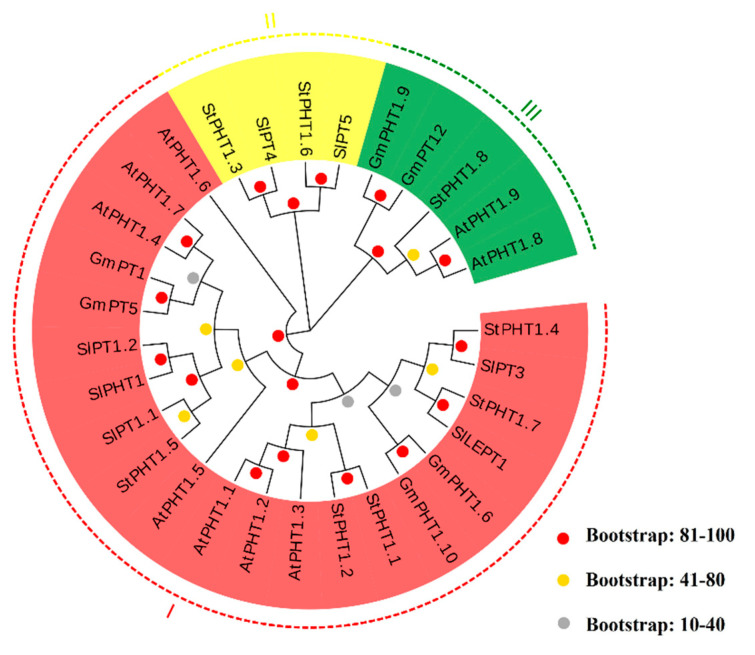
The phylogenetic tree of PHT1 proteins of potato, tomato, *Arabidopsis*, and soybean. The phylogenetic tree was constructed using the neighbor-joining method with 1000 bootstrap replications. PHT1 genes were distributed in three main groups, including I, II, and III, which were marked with different colors. PHT1 families are indicated with different colors. Prefx “St” indicates *Solanum tuberosum* (potato), “Gm” indicates *Glycine max* (soybean), “At” indicates *Arabidopsis thaliana*, and “Sl” indicates *Solanum lycopersicum* (tomato). At the nodes, red dots represent bootstrap values that were between 81 and 100; yellow dots show the bootstrap values that were between 41 and 80; and gray dots indicate that the bootstrap values were less than or equal to 40.

**Figure 4 plants-09-01384-f004:**
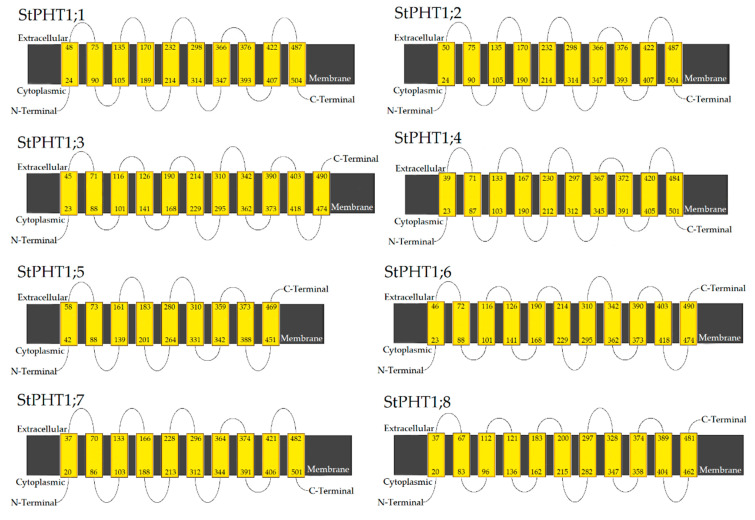
Prediction of the transmembrane domain of the potato *PHT1* gene family. The numbers at the top and bottom of the yellow boxes indicate where the polypeptide chain is transmembrane.

**Figure 5 plants-09-01384-f005:**
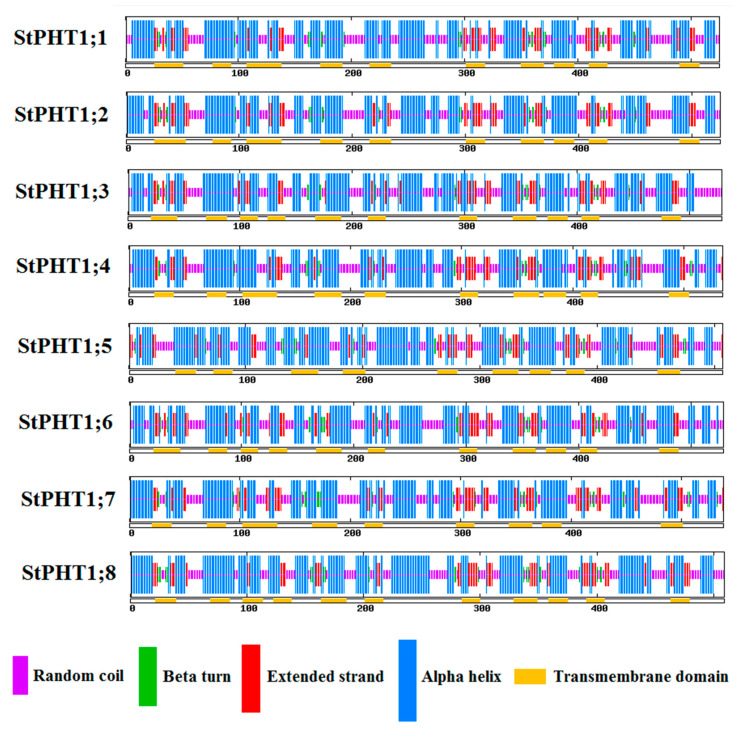
Analysis of the secondary structures of potato *PHT1* gene family proteins. Purple boxes indicate random coil, green indicate beta turn, red indicate extended strand, and blue indicate alpha helix. The yellow boxes under each secondary structure box indicate transmembrane domain.

**Figure 6 plants-09-01384-f006:**
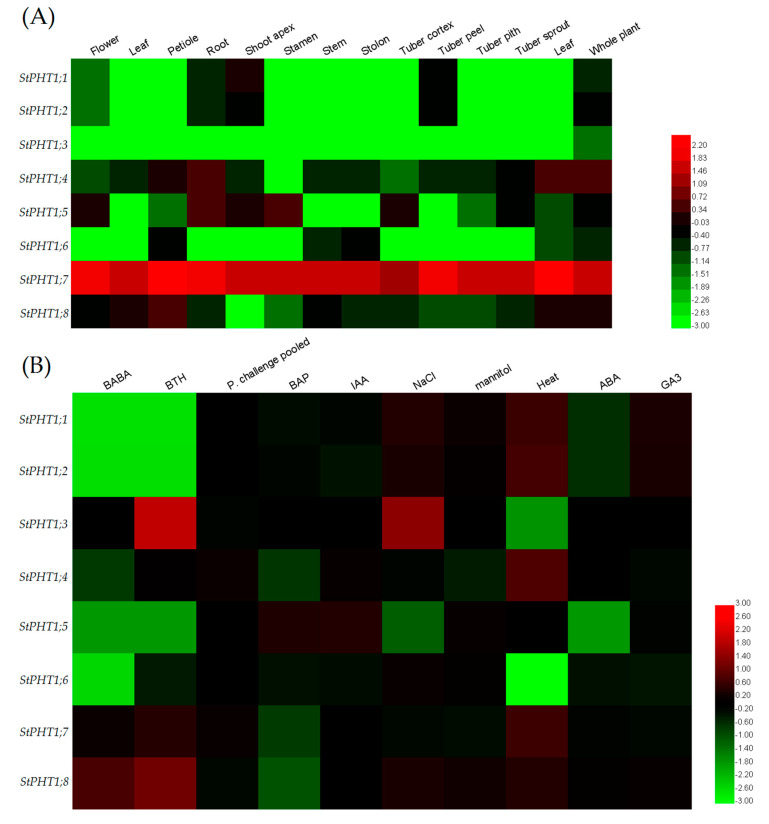
(**A**) Heat map of potato *PHT1* genes tissue-specific expression. The data used in the figure was the base-10 logarithm of raw data from the PGSC database. Transcripts were detected using RNA-seq technology. Red indicates high relative gene expression, whereas green indicates low relative gene expression. (**B**) Heat map of the potato *PHT1* genes under different stresses and phytohormone treatments. The data used in the figure = lg (experimental value/control value). The experimental value and control value were obtained from the PGSC database. Transcripts were detected using RNA-seq technology. Abiotic stresses included salt, mannitol, and heat; biotic stresses included DL-b-amino-n-butyric acid (BABA), stress-elicitors acibenzolar-*S*-methyl (BTH), and *Phytophthora infestans*; and other stress responses were mainly induced by four plant hormones: 6-benzylaminopurine (BAP), indole-3-acetic acid (IAA), abscisic acid (ABA), and gibberellic acid (GA3). Red indicates gene up-regulation, while green indicates gene down-regulation.

**Figure 7 plants-09-01384-f007:**
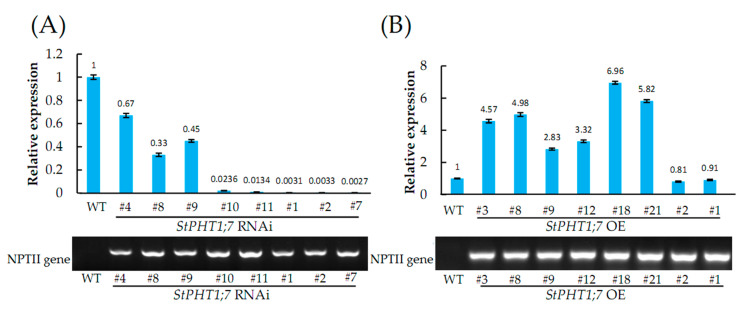
PCR and qRT-PCR analysis of *StPHT1;7* silenced (RNAi) (**A**) and overexpressed (OE) (**B**) plants. Upper: relative expression of *StPHT1;7*; bottom: identification of *NPTII*. Each value is expressed as mean ± SD (*n* = 9,3 biological repeats × 3 technical repeats). PCR and qRT-PCR samples were taken from the corresponding *in vitro* potato plantlets cultured for 30 days.

**Figure 8 plants-09-01384-f008:**
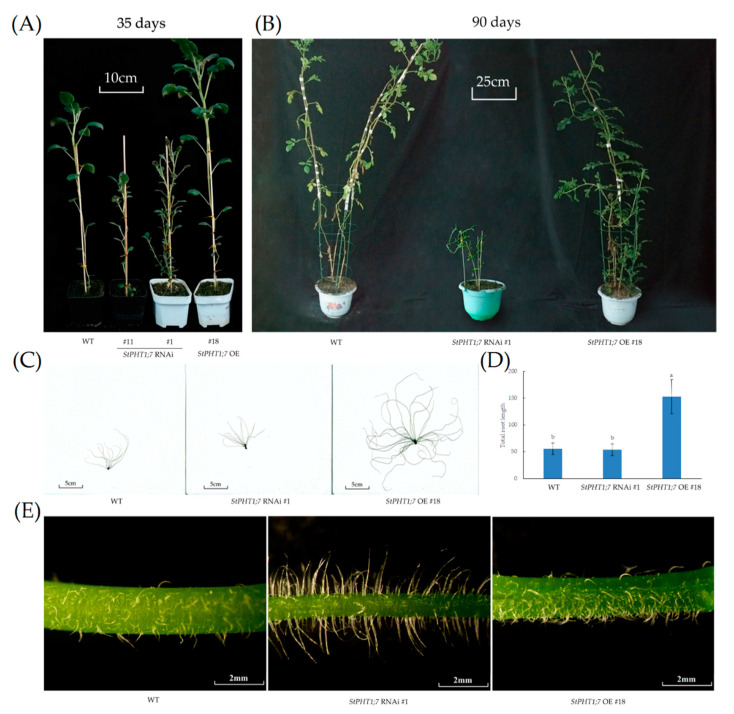
(**A**) Phenotype of negative control (WT), *StPHT1;7* RNAi #1, *StPHT1;7* RNAi #11, and *StPHT1;7* OE #18 transgenic potato plants 35 days after transplanting. (**B**) Phenotypes of WT, *StPHT1;7* RNAi #1 and *StPHT1;7* OE #18 transgenic potato plants 3 months after transplanting. (**C**) Root scanning of WT, *StPHT1;7* RNAi #1, and *StPHT1;7* OE #18 transgenic tissue culture seedlings in the MS2 liquid culture medium. (**D**) Total root length of WT, *StPHT1;7* RNAi #1, and *StPHT1;7* OE #18 transgenic tissue culture seedlings. Each value is expressed as mean ± SD (*n* = 5, 5 biological repeats). Different lowercase letter represents significant differences (*p* ≤ 0.05) using Duncan’s multiple range test, respectively. (**E**) Stem hairs of WT, *StPHT1;7* RNAi #1, and *StPHT1;7* OE #18 transgenic potato lines 30 days after Hoagland hydroponics.

**Figure 9 plants-09-01384-f009:**
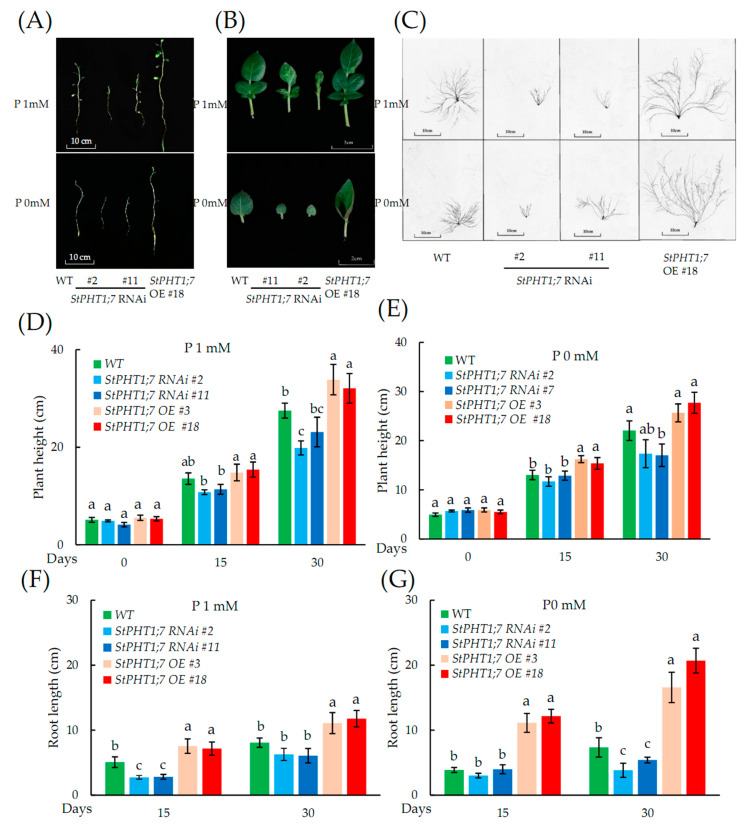
Changes in plant height, root length, and leaf shape among different transgenic potato lines under phosphorus-free conditions. “P1mM” indicates normal P content (1 mM) cultivation, while “P0mM” indicates phosphorus-free content (0 mM) cultivation. (**A**) Phenotypes of potato plants in Hoagland hydroponics at 30 days; (**B**) the phenotype of the third fully expanded leaves of potato plants under the conditions of Hoagland hydroponics at 30 days; (**C**) roots of different transgenic potato lines at 30 days of P stress; (**D**) the plant height of different transgenic potato lines under normal P content (1 mM) cultivation; (**E**) the plant height of different transgenic potato lines during P-free stress cultivation; (**F**) the root length of different transgenic potato lines during normal P content (1 mM) cultivation; (**G**) the root length of different transgenic potato lines during P-free stress cultivation. Each value is expressed as mean ± SD (*n* = 9, 9 biological repeats). Different lowercase letters in the same column represent significant differences (*p* ≤ 0.05) using Duncan’s multiple range test, respectively.

**Figure 10 plants-09-01384-f010:**
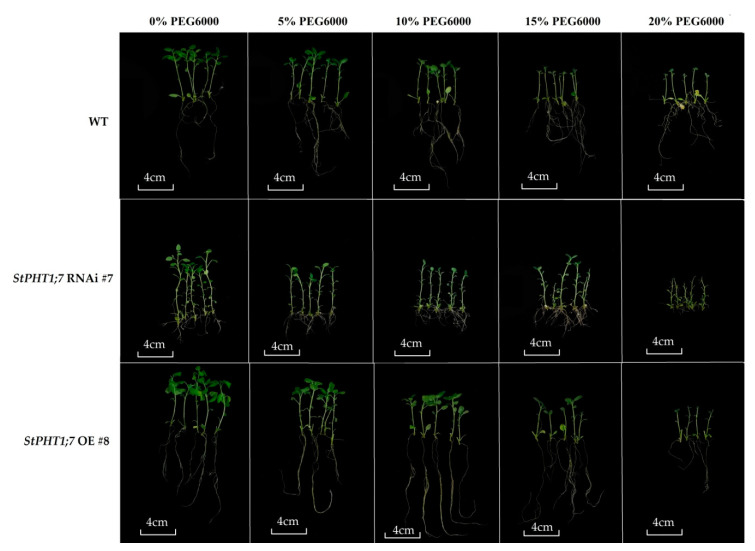
Morphological characteristics of transgenic *in vitro* potato plantlets under (Polyethylene glycol) PEG treatments. The transgenic *in vitro* potato plantlets grown for 20 days were transplanted into MS solid medium containing different concentrations of PEG and treated for 25 days.

**Figure 11 plants-09-01384-f011:**
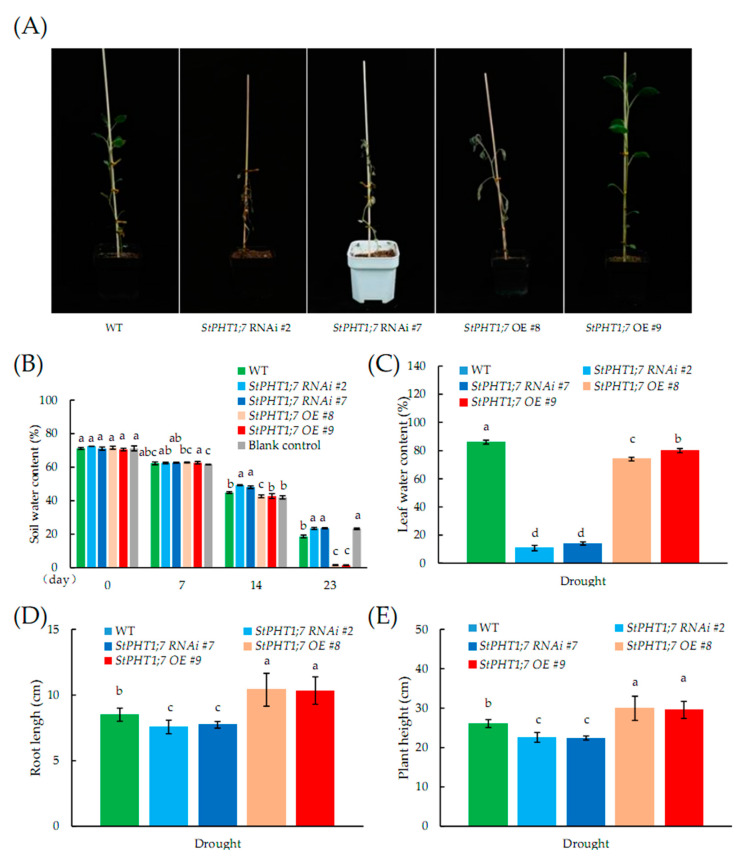
(**A**) Growth phenotype of different transgenic potato plants after 23 days of natural drought; (**B**) soil water content during the trial; (**C**) leaf water content of different transgenic potato lines after 23 days of drought treatment; (**D**) the root length of different transgenic potato lines after 23 days of drought treatment; (**E**) the plant height of different transgenic potato lines after 23 days of drought treatment. Each value is expressed as mean ± SD (*n* = 9,3 biological repeats × 3 technical repeats). Different lowercase letters in the same column represent significant differences (*p* ≤ 0.05) using Duncan’s multiple range test, respectively.

**Table 1 plants-09-01384-t001:** Multiple comparisons of the relative growth indices under 15% PEG6000-induced drought stress of the in vitro potato plantlets of five lines.

Line	Growth Parameter (%)						
	Relative Plant Height	Relative Stem Fresh Weight	Relative Stem Dry Weight	Relative Root Length	Relative Root Fresh Weight	Relative Root Dry Weight	Relative Leaf Area
WT	70.21a	44.84a	75.57a	58.65b	127.58a	193.75a	46.14a
*StPHT1;7* RNAi #2	62.90a	38.24b	55.15b	94.12ab	81.80b	113.16b	35.25b
*StPHT1;7* RNAi #7	57.40b	49.44a	69.78a	94.94ab	105.20b	119.66b	49.28a
*StPHT1;7* OE #21	53.54b	31.17b	44.10b	53.38b	106.97b	129.63b	17.56c
*StPHT1;7* OE #8	62.51a	43.81a	73.35a	123.09a	84.61b	103.77b	51.04a
Mean	61.31	41.50	63.59	84.83	101.23	131.99	39.85
SD	10.23	13.87	20.13	38.58	49.06	75.97	26.13
CV	16.68	33.42	31..66	45.48	48.46	57.55	65.57

Different lowercase letters in the same column represent significant differences (*p* ≤ 0.05) using Duncan’s multiple range test, respectively.

**Table 2 plants-09-01384-t002:** Media used for potato transformation.

Medium	pH	Composition
Solid MS2	5.8	Murashige and Skoog (MS) powder 4.42 g/L; sucrose: 20 g/L; agar: 8 g/L
Liquid MS2	5.8	MS powder 4.42 g/L; sucrose: 20 g/L
CIM	5.8	MS powder 4.42 g/L; glucose: 1.6 g/L; MES: 0.5 g/L; agar: 6.5 g/L; 6-benzylaminopurine (6-BA): 0.1 mg/L; 1-naphthylacetic acid (NAA): 5 mg/L; kanamycin (Kan): 50 mg/L; timentin (TM): 200 mg/L
SIM	5.8	MS powder 4.42 g/L; glucose: 1.6 g/L; MES: 0.5 g/L; agar: 6.5 g/L; gibberellin A3 (GA_3_): 0.1 mg/L; NAA: 0.02 mg/L; zeatin (ZT): 2 mg/L; Kan: 50 mg/L; TM: 200 mg/L
SM	5.8	MS powder 4.42 g/L; sucrose: 20 g/L; agar: 8 g/L; Kan: 50 mg/L

Abbreviations: callus induced medium (CIM); shoot induced medium (SIM); selected medium (SM).

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
