# Peer review of "Functional Analysis of StPHT1;7, a Solanum tuberosum L. Phosphate Transporter Gene, in Growth and Drought Tolerance"

_plants, 2020, doi:10.3390/plants9101384_

Round 1

Reviewer 1 Report

The manuscript characterized the structure and function of PHT1;7 in the response of potatoes to drought stress. Although the manuscript is interesting, many issues should be addressed to strengthen the conclusions and to improve the manuscript.

  1. Figure 9 and 10 are the most important data for the manuscript. However, the data presented and the description of the results are not clear. For example, the seedling and root growth of the OE #8 in Figure 9 seems better than those of wild type under 0%, 5%, 10%, and 15% PEG. The growth parameter (% relative value) in Table 1 is hard to understand. In addition, drought stress-responsive phenotypes in soil (Figure 10) should be described in more detail, including plant height, fresh weight, root length, leaf area, etc. Without these parameters, drought tolerance of OE and RNAi lines cannot be decided precisely.
  2. In abstract, PHT1 should be defined. The conclusion that the tolerance to drought stress was ranked as wild type > overexpression line > silence line should be decided with caution after more in-depth analysis of OE and RNAi lines under drought stress (see below).
  3. Figure 1 legend; it would be more informative if the specific features of each motif are briefly described.
  4. Line 113, the subtitle 2.1. Gene Structure and Conservative Motif Analysis of the Potato PHT1 Gene Family should be Gene Structure and Conservative Motifs of the Potato PHT1 Gene Family.
  5. Line 124, the subtitle 2.2. Construction of Potato PHT1 Phylogenetic Tree should be The phylogenetic tree of Potato PHT1 genes.
  6. Line 141, the subtitle 2.3. Potato PHT1 Protein located on the cell membrane should be Potato PHT1 Protein is located on the cell membrane.
  7. Figure 3 legend; it would be more informative if the meaning of numbers is explained.
  8. Figure 4; it would be more informative if the positions of transmembrane domains shown in Figure 3 are indicated above the secondary structure.
  9. Figure 6 legend; the experimental conditions for PCR and qRT-PCR, such as timing of sample collection, should be described.
  10. Figure 7; OE #18 is shown on the figure, but the figure legend says OE #21.
  11. Line 237, the subtitle 2.8. Transgenic Plant Phosphorus Stress Treatment should be Responses of the Transgenic Plant to Phosphorus Stress.
  12. Figure 8F and 8G; it would be better to have the y-axis same scale to compare easily the root length between P 1 mM and P 0 mM.
  13. Line 286, the subtitle 2.9. Transgenic Plant PEG6000 Stress Treatment should be Response of the Transgenic Plants to Dehydration Stress.
  14. Figure 9 legend; the experimental conditions, such as timing of PEG treatment, duration of PEG treatment, etc., should be described.
  15. Figure 10; more pictures of other OE and RNAi lines should be shown.
  16. Many part of the Discussion is just a repetition of the Results, which should be revised to emphasize the meaning or implication of current results and to compare the current results with the previously known facts. Subtitles in Discussion also should be revised to emphasize or point out the main conclusion in each section.

Reviewer 2 Report

Dear authors,   The paper submitted by Cao et al. describes the functional analysis of a phosphate transporter gene (StPHT1;7) in growth and drough tolerance in Solanum tuberosum L. Overall, the presented methodology is solid and the topic is scientifically interesting. However, some minor issues need to be addressed.   

  • L101: “climate change has become unstable” Statement is unclear. Do you mean “climate has become unstable?”
  • L114: The assembly/annotation version used should be in the main text
  • L156 and 161: There are two table 2 in the main text. I suggest putting this table 2 into the suplementent
  • The analysis of the gene structure and motifs is largely descriptive and mostly refers to supplementary data. I suggest combining all structural analyses and rewrite it as one paragraph. Additionally, a multiple sequence alignment should be included.
  • L494: No space between XbaI and SacI
  • Version numbers of the software used should be included in the methods

Round 2

Reviewer 1 Report

The authors addressed all of my comments and suggestion, which greatly clarified and improved the manuscript. I have no further comments.

This manuscript is a resubmission of an earlier submission. The following is a list of the peer review reports and author responses from that submission.

Round 1

Reviewer 1 Report

Dear authors,

Here are my suggestions. You should be particularly careful with the "scientific language", statistical analysis and experiment design description.

Abstract

Line 19: the authors should show also relative expression in fold instead of changing to %

Introduction

Line 28 "is one of the important biological macromolecules in plants.."

Line 38: "root length are reduced.."

Line 40: "root length is shortened" the authors already refer this in line 38

Line 42: please rewrite this sentence, my suggestion "in the form of inorganic phosphorous which has a low prevalence in soils"

Line 71: remove the word genes from "The protein sequence of potato PHT1 family genes"

Line 72-73: "and all amino acids have an isoelectric point greater than eight" this is odd. So the PHT1 family proteins do not have any aspartic/glutamic acid, glutamine, serine, cysteine, etc residues? Or do you mean the total protein pI? It would be helpful to have NCBI codes of the family proteins

Line 81: "turn the leaves yellow"

Line 98: I think there's an H missing in OSPT1, comparing to the sentence in line 86

Results and Methods

Fig.1: Should separate the figures in panels. GSDS image is not good qaulity. right hand side of domain scheme is too small, I cannot read the information regarding E-values of domains. In the figure legend you should say which is which, meaning structure analysis (panel A) domain prediction (panel B) or something similar

Line 126: "the predicted transmembrane domains..."

Line 127-28: "N- terminus is in the cytoplasm"

Fig2. Poor quality image. Please correct

Line 148 to 160: How do you select "categories"? This is usually referred to as clusters, and to do so you should only compare the proteins from potato. You can present one phylogenetic tree for PHT1 genes from potato and another one where you add them to the the ones from At and Gm. Also this should be referred in the figure legend; fig 4 shows more than the phylogenetic tree of potato proteins. You should also add in the methods section which algorithm you used in MEGA, they have at least 3

Fig 5. In the legend, you have to say that this is based on the PGSC database, these are not your results

Lines 173-182: what threshold do you consider for decrease/increase? For instances, you say that under salt stress expression of PHT1;4,  1;5 and 1;7 decrease while it increases for the remaining genes. However the color for 1;4 is the same for 1;6; what is the criteria?

Fig 6. graphs on this figure are of poor quality, please improve them, and add Relative expression of StPHT1;7. Please add next to the PCR slice the name of the gene you are showing

Line 203 to 217: why don't you refer to fold-change? This would be clearer to me if in the methods section you describe the calculations you've done. You should indicate how you have done the RNA extraction, cDNA synthesis and how much RNA you used; how did you confirm that you do not have gDNA contamination; which primers did you use in cDNA synthesis; negative control; biological and technical replicates; what statistical analysis have you done;

the authors must use the most recent equation to calculate the expression variation of the genes analysed - Hellemans et al., 2007 DOI:10.1186/gb-2007-8-2-r19. This equation considers amplification efficiency for each run and used a normalization factor (NF) based on multiple reference genes for expression correction. OAfter qPCR data analysis using Hellemans method, a column with the primer’s amplification efficiency must be included in a table.

Section 2.7: I understand why you chose RNAi #1 and #11; why not #10, #2 or #7? why OE line #21 on figure 7 panel A? You should say why you show some in figure 7 and not others.

Figure 7 panel A and B do not show the same lines; for example, I cannot compare line #21 OE at 35 and 90 days; the lines should be the same so a comparison can be made. 

Fig 7. The transgenic lines in figure 7 should be the same in all the panels; you should also refer the transgenic lines in the figure legend.

Line 235: "while the root system of the silence plant was slightly weaker than that of the WT" this is not shown in the graph on Fig.7 D, they are actually very similar

Line 248 to 249: "Similarly, here, the indicators of the StPHT1;7 RNAi #11 line are better than those of StPHT1;7 RNAi #2 and are closer to the WT." what does better mean here? what indicators?

Fig 8: add the legend of P1 and P0 for panels A and B; error bar legend for panels D to G is missing (SD? n=?)

Line 281-282: "Compared with the WT, the overexpression line was significantly higher." You shouldn't say this, significant how? You do not show any statistical analysis; what are the error bars? there is no reference to a statistical analysis in your methods section or in the figure legend. It does not appear to be significant since error bars coincide, significance usually refers to a ANOVA test or similar, but please verify

Line 305: "supp table 1 shows the growth indexes of ..."

Line 307: " Most of the indexes of the line were..." which line?

Line 311: "significantly different..." again you cannot state this when you do not perform statistical analysis; you do not refer statistical analysis in the methods section

Line 315: " the roots of the other four lines " which ones?

Table 2: refer in the methods section the statistical analysis

Methods

Section 4. 8 You have to refer how did you transform Agrobacterium and how did you transform the plants: media; growth conditions; transformation method...

Section 4.10: what are the greenhouse conditions; how many replicates?

Reviewer 2 Report

The manuscript entitled “Functional Analysis of StPHT1;7, Solanum tuberosum L. Phosphate Transporter Related Gene in Growth and Drought Tolerance” illustrates an effort towards identification of PHT1 gene family in potato and functional analysis of the StPHT1-7 gene,  especially in view of its role in drought tolerance. Overall, the manuscript can be considered for publication after major and minor revisions aiming to improve the clarity of the manuscript.

General remark

In my opinion, for abiotic stresses, the terms stress tolerance and stress resistance should not be used as synonyms. Here, I suggest  to use the term stress tolerance in whole ms. 

The specific comments are as follow:

Results:

Paragraph 2.5:

Procedures used for the stress treatments presented in Fig 5B are not described in M&M. It should be corrected.

I am not convinced that using of the terms “maturity tubers” and “young tubers” in Fig 5A is adequate in the context of performed experiments. 

Paragraph 2.6:

lane 206: what about expression of silence line #10?, please explain

lane 210: … highest is 6.96 times for the line SPHT1;7 RNAi #18….; please correct

lane 213: “The overexpression line StPHT1;7 was…” ; I guess that it should be “The overexpression lines StPHT1;7 were” 

Paragraph 2.7:

lane 237: “reaching 2.74 times”, please add for which line

lane 242: “The WT and overexpression lines were also different to a certain extent” The sentence is not informative. 

Paragraph 2.8:

Figure 8A is totally unreadable.The caption of  Figure 8 need to be edited. “P1 and P0” on Figs 8A-C: signs  should be explained.

lanes 278-296: comparison between two overexpression lines and two silence lines was done. Therefore, the plural form should be used in the description. 

Paragraph 2.9:

There is a conclusion at the end of this section (lanes 407-409): “This shows that the WT was less affected by PEG6000 stress and had strong drought tolerance. Each of the remaining four lines had multiple indicators that were lower or significantly lower than those in the other lines, so the drought resistance of the remaining four lines could not be judged here”. I read this statement that materials or PEG6000 stress used in the experiments do not provide new data to better understand potato response to drought. Therefore, this paragraph should be prepared in much more compact version, without description in details data described in supplementary tables. Figure 9 is not informative. 

Paragraph 2.10:

Supplementary data should not be a basis for result description. Figure 10 is not readable and informative. I suggest to show the table data in the main text. The caption of Supplemental Table 3 need to be edited.

Discussion

“The results of the PEG treatment in this experiment were roughly the same” When reading this sentence and whole discussion, it is difficult to find the innovative meaning of the results obtained.

MS is submitted to Agronomy journal. It is well known that different stress-responses in potato cultivars at the leaf or stem level do not have to affect into tuber yield. For breeding and potato industry, differences in tuber yield should be taken as a main criterion of potato tolerance to abiotic stress. Therefore, it is advisable to include in discussion a conclusion about the potential significance of the presented data for potato breeding.